# Chitosan Biocomposites with Variable Cross-Linking and Copper-Doping for Enhanced Phosphate Removal

**DOI:** 10.3390/molecules29020445

**Published:** 2024-01-16

**Authors:** Inimfon A. Udoetok, Abdalla H. Karoyo, Mohamed H. Mohamed, Lee D. Wilson

**Affiliations:** Department of Chemistry, University of Saskatchewan, 110 Science Place, Saskatoon, SK S7N 5C9, Canadaabk726@mail.usask.ca (A.H.K.);

**Keywords:** chitosan beads, biocomposite adsorbent, glutaraldehyde cross-linking, Cu(II) ion complexation, phosphate

## Abstract

The fabrication of chitosan (CH) biocomposite beads with variable copper (Cu^2+^) ion doping was achieved with a glutaraldehyde cross-linker (CL) through three distinct methods: (1) formation of CH beads was followed by imbibition of Cu(II) ions (CH-*b*-Cu) without CL; (2) cross-linking of the CH beads, followed by imbibition of Cu(II) ions (CH-*b*-CL-Cu); and (3) cross-linking of pristine CH, followed by bead formation with Cu(II) imbibing onto the beads (CH-CL-*b*-Cu). The biocomposites (CH-*b*-Cu, CH-*b*-CL-Cu, and CH-CL-*b*-Cu) were characterized via spectroscopy (FTIR, ^13^C solid NMR, XPS), SEM, TGA, equilibrium solvent swelling methods, and phosphate adsorption isotherms. The results reveal variable cross-linking and Cu(II) doping of the CH beads, in accordance with the step-wise design strategy. CH-CL-*b*-Cu exhibited the greatest pillaring of chitosan fibrils with greater cross-linking, along with low Cu(II) loading, reduced solvent swelling, and attenuated uptake of phosphate dianions. Equilibrium and kinetic uptake results at pH 8.5 and 295 K reveal that the non-CL Cu-imbibed beads (CH-*b*-Cu) display the highest affinity for phosphate (Q*_m_* = 133 ± 45 mg/g), in agreement with the highest loading of Cu(II) and enhanced water swelling. Regeneration studies demonstrated the sustainability and cost-effectiveness of Cu-imbibed chitosan beads for controlled phosphate removal, whilst maintaining over 80% regenerability across several adsorption–desorption cycles. This study offers a facile synthetic approach for controlled Cu^2+^ ion doping onto chitosan-based beads, enabling tailored phosphate oxyanion uptake from aqueous media by employing a sustainable polysaccharide biocomposite adsorbent for water remediation by mitigation of eutrophication.

## 1. Introduction

Eutrophication is the accumulation of growth factors like carbon dioxide, and nutrient salts (e.g., phosphorus and nitrogen) in water bodies [1,2]. The increased rate and extent of eutrophication of water bodies relate to elevated levels of nutrients in aquatic ecosystems [3,4], such as phosphate and nitrate, which serve to increase the productivity and density of aquatic vegetation and fishes for recreational and economic purposes [5]. Eutrophication of water bodies may also be linked to anthropogenic activities such as agriculture, industrialization, sewage disposal, and fossil fuel emissions [6,7,8]. In turn, an increased production level of algae and other aquatic plants, reduced oxygen levels (hypoxia), reduced biodiversity due to the death of aquatic life, and a reduction in the general quality of water can occur [9]. Severe eutrophication significantly alters ecosystem health and poses challenges to addressing global water security, which highlights an urgent need to develop sustainable and cost-effective technologies to address such challenges in aquatic environments [10].

Various methods have been employed to mitigate eutrophication, such as the use of herbicides and algaecides, physical agitation of the water to homogenize nutrients, temperature and oxygen distribution, and diversion of excess nutrients by physical, chemical or biological methods [11,12,13]. Among the various methods available, [14,15,16] adsorption has distinct advantages over the other approaches since it is technically feasible, cost-effective, and environmentally friendly [9,13,14,17,18]. A key challenge for emerging adsorption technology relates to the development of effective and sustainable adsorbents [15,16].

Chitosan (CH) is an abundant cationic polysaccharide with primary amine (-NH_2_) and hydroxy (-OH) groups [19], which can undergo modification at acidic and alkaline conditions to achieve functional materials with improved adsorption properties [20]. The unique physicochemical properties of chitosan have led to the preparation of cross-linked powders [21], flakes [20], beads [22], and biocomposite adsorbents [23,24,25,26] for environmental remediation. Cross-linking of chitosan using low molecular weight cross-linker agents (e.g., glutaraldehyde, epichlorohydrin, and citric acid) presents a versatile synthetic strategy that affords modified materials with tunable surface and textural properties for targeted sequestration of diverse waterborne contaminants [20,22]. The variable affinity of chitosan-based adsorbents for phosphate relates to the presence of available NH_2_ groups, where synthetic modification can enhance the mechanical stability, porosity, and interfacial interaction of the sorbent material with inorganic phosphate dianions (P_i_). Chitosan-based adsorbents in the form of native or cross-linked chitosan [27,28,29], where the bead systems are either cross-linked and/or imbibed with metal ions [29,30,31], quaternized beads [32,33], and hybrid biocomposites [17,23,34], have been applied as adsorbents for the removal of waterborne P_i_. In particular, chitosan beads have received ongoing research attention because they are amenable to modular synthesis that contributes to their ease of use as adsorbents for controlled removal of contaminants (e.g., filter columns, cartridges, etc.). Various studies [35,36] have demonstrated that among divalent cations, copper is considered one of the most versatile metals with chitosan. In particular, Wu et al. [37] reported a Cu^2+^-crosslinked chitosan (chitosan–Cu) material as a stable and high-performance hydroxide exchange membranes (HEMs), where the prepared materials were able to accommodate water diffusion and facilitate rapid ion transport. Copper was selected as the metal component of the composite in the current study because there are studies that report on the preparation of copper–chitosan composites. By comparison, systematic studies that outline the effects of chitosan cross-linking, bead formation, and metal ion imbibition are sparsely reported [38,39,40,41]. Herein, we describe the preparation and characterization of copper (Cu)-imbibed chitosan-based adsorbents that employ glutaraldehyde as the cross-linking agent, according to a step-wise synthesis (cf. Figure 1a). The formation of beads employed pristine and cross-linked chitosan that affords materials with variable mechanical, textural, and water swelling properties (cf. Figure 1b) [42]. The modified bead materials were characterized using spectroscopy (FTIR, NMR, and XPS), SEM, thermoanalytical (TGA), and sorption-based methods (water swelling and phosphate uptake isotherms). In turn, this study outlines several contributions that relate to advanced polysaccharide materials and their physicochemical properties, as follows: (i) the development of a greater understanding of the role of cross-linking and bead formation on the adsorbent structure, (ii) the effect of glutaraldehyde cross-linking and the role of Cu imbibition, and (iii) insight on the structure–adsorption relationships of the adsorbents obtained via controlled Cu-doping. This study represents the first example that reports on the role of the steps in the modular synthesis of biocomposites, which results from the cross-linking of chitosan with glutaraldehyde, phase inversion synthesis, and metal imbibing, before and after bead formation. Furthermore, the structure–property relationships of the biocomposite adsorbents are evaluated herein for the removal of phosphate dianions in aqueous media.

## 2. Results and Discussion

An underlying hypothesis in this study posits that cross-linking of chitosan, Cu(II) complexation, and bead formation yield stable and sustainable adsorbents for the controlled removal of phosphate dianions from aqueous media. Moreover, the sequence of synthetic steps (cf. Figure 1a) is anticipated to play a role in modifying the adsorption properties of such chitosan bead systems. The bead systems were structurally characterized by TGA and spectral (IR, XPS, and NMR) analysis, whereas the sorption properties included evaluation of the water swelling and adsorption of phosphate at equilibrium and kinetic conditions, as described in Section 2.

### 2.1. Characterization Studies

#### 2.1.1. FTIR Spectroscopy

FTIR is a sensitive spectral technique for studying the surface functionality of chitosan beads upon cross-linking and/or imbibition of Cu(II) ions. Structural characterization of chitosan using FTIR is complicated due to the contribution of similar frequencies of the functional groups for both chitosan and its cross-linked forms with glutaraldehyde. However, prominent vibrational bands are shown in Figure 1 for pristine chitosan beads (CH-*b*). The band at ~3000–3500 cm^−1^ is associated with the stretching vibrations of the O-H and N-H groups of the chitosan backbone. The IR band near ~2900 cm^−1^ corresponds to the aliphatic -CH asymmetric stretching of chitosan. The bands at ~1650 cm^−1^ and ~1590 cm^−1^ are assigned to the amide I (C=O) of the remaining acetamide group of the deacetylated (85%) chitosan and the NH_2_ bending (δNH_2_) frequency, respectively. Other bands are noted at ~1000–1200 cm^−1^ (C–O–H, C–O–C, and C–N–H stretching) and below ~700 cm^−1^ (chitosan skeletal deformation) [43]. The cross-linking of the chitosan biopolymer and/or coordination of Cu ions is supported by the attenuated δNH_2_ band at ~1590 cm^−1^, (marked with * in Figure 1) followed by the formation of an imine (ν_as_C=N) group that is overlapped with the amide I band at ~1650 cm^−1^. The band at ~1650 cm^−1^ for the imine group becomes broader with red shifts (ca.−15 cm^−1^) for the cross-linked chitosan beads (CH-CL-*b*-Cu and CH-*b*-CL-Cu ~1649 cm^−1^) relative to the pristine chitosan beads (CH-*b* ~1664 cm^−1^). The latter provides further support for the glutaraldehyde cross-linking of chitosan, according to the Schiff base reaction mechanism [44]. The broadened and reduced intensity of the band related to the O-H and N-H groups at ~3400 cm^−1^ indicate that the primary NH_2_ groups are involved in the cross-linking process [45].

The incremental spectral shifts of the NH_2_ band at ~1590 cm^−1^ (marked with * in Figure 1) in the Cu-doped chitosan beads provide support that chitosan–Cu(II) complexes are formed. In particular, the attenuation of the C–H stretching vibration bands near 2900 cm^−1^ and the emergence of a new band at 600 cm^−1^ in the spectra of CH-*b*-Cu, CH-CL-*b*-Cu, and CH-*b*-CL-Cu samples further supports that interaction occurs between chitosan and Cu in agreement with previous reports [31,46]. The C-O band at ~1150 cm^−1^ (marked with * in Figure 1) is prominent and was used to support the coordination of Cu(II) ions with chitosan, in line with the “bridge” model [47]. In general, the degree of Cu loading and formation of bridge- versus pendant-coordinated Cu complexes depend on the availability of surface-accessible active sites, pore structure, and the cross-linking density of the beads. The bridge model is anticipated for all three samples, whereas the pendant model may be more prominent in the CH-CL-*b*-Cu sample due to extensive cross-linking and the lack of sufficient active sites on the beads. The greater intensity and broad O-H band for the CH-CL-*b*-Cu indicates that a pillaring effect results from extensive cross-linking, which affords a greater number of free O-H groups. However, the disparity in the hydration of the samples is supported by much broader O-H bands for the CH-*b*-Cu and CH-*b*-CL-Cu materials. This includes the variable DTG profiles in Section 2.1.2 that parallel variable Cu loading and the presence of a heterogeneous microenvironment in these biocomposites. Furthermore, the appearance of the Cu-O stretching band at ~613 cm^−1^ for CH-*b*-CL-Cu, relative to ~623 cm^−1^ for CH-*b*-Cu and CH-CL-*b*-Cu [41], may indicate a Cu microenvironment that differs from the CH-CL-*b*-Cu sample, as described above.

#### 2.1.2. Thermal Gravimetry Analysis (TGA)

The variation in thermal stability of the chitosan beads was evaluated using TGA as shown in Figure 2. The TGA results of the Cu-imbibed chitosan-based materials are presented as plots of derivative weight (%/°C) versus temperature (°C) and compared with pristine chitosan beads (CH-b). The thermogram for pristine CH-b reveals two main thermal events: a weak dehydration enthalpy (∆H_deh_) event covering a wide temperature range (~100–250 °C) and an intense decomposition peak at ~300 °C. The weak and broader ∆H_deh_ for the CH-b sample indicates a variable range of bound water at the surface sites and micropore structure of the beads, in the absence of doped Cu ions. Similar dehydration features are shown for the Cu-imbibed chitosan biocomposite beads (CH-CL-b-Cu), albeit at a much lower temperature (~75 °C). This indicates a lack of accessible active sites for chitosan, glutaraldehyde, and/or interstitial regions due to extensive cross-linking. The latter may account for low Cu doping in this sample and the concomitantly lower water uptake properties, in agreement with the FTIR results. By contrast, the non-cross-linked- (CH-b-Cu; ~75 °C) and surface cross-linked- (CH-b-CL-Cu; ~110 °C) Cu-imbibed chitosan beads reveal relatively higher ∆H_deh_ at much lower temperatures. Abundant Cu sites and other heteroatom functional groups (NH_2_, C=O, O-H) due to chitosan and partially cross-linked glutaraldehyde may enhance Cu doping and water binding in these samples, in agreement with FTIR spectral results. The lower degradation temperature (~225 °C) for the non-cross-linked and cross-linked chitosan beads is attributed to the effects of cross-linking and/or Cu doping, in agreement with other reports [48,49]. Cross-linking/Cu doping of the chitosan beads is anticipated to attenuate the inter-/intramolecular hydrogen bonding in the chitosan biopolymer units that result in a modified heat capacity [49]. The attenuation of hydrogen bonding in the biopolymer matrix is supported by the decrease in the intensity and broadening of the O-H and N-H stretching bands (~3000–3500 cm^−1^) in the FTIR spectra. The formation of variable (pendant- and/or bridge-coordinated) Cu^2+^ complexes and possible structural defects in the chitosan biopolymer may relate to variable thermal degradation temperatures with a consequent decrease in the heat capacity. Ng et al. [50] also reported a decrease in the DSC decomposition temperature of a chitosan–copper complex that concurs with the TGA results reported in this study.

#### 2.1.3. Solid State ^13^C NMR Spectroscopy

The structural changes in the beads due to cross-linking with glutaraldehyde and/or imbibition of Cu species were studied through solid-state NMR spectroscopy. The spectra of the chitosan beads in Figure 3 were assigned from a previous report [23]. The NMR spectra display relatively well-resolved ^13^C resonance signals at ~105 ppm (C-1), ~56.8 ppm (C-2), ~75.0 ppm (C-3, C-5), ~82.4 ppm (C-4), ~60.6 ppm (C-6), ~174 ppm (C=O), and ~23.1 ppm (CH_3_). The broader signals for the CH-*b*-Cu, CH-CL-*b*-Cu, and CH-*b*-CL-Cu samples relate to the effects of cross-linking and/or complex formation with Cu(II) species within the biocomposite, in agreement with the FTIR results. The reduced resolution for the Cu-doped/cross-linked chitosan bead materials is attributed to the crystallographic inequivalence of the glucosamine units for the powdered samples with variable hydration states, which also show parallel agreement with the TGA and FTIR results. Cross-linking and coordination of Cu(II) ions with chitosan is anticipated to reduce the structural and dynamic flexibility of the biopolymer, in agreement with results for calcium-doped chitosan beads [42] and other cross-linked chitosan materials [27,51,52].

#### 2.1.4. Scanning Electron Microscopy (SEM)

The variation of the morphological and surface properties of the beads were examined through analysis of SEM images, as shown in Figure 4. The CH-*b* exhibits a homogenous microporous morphology with a very compact and smooth surface. Similar morphological features are shown for the CH-CL-*b*-Cu sample with a slightly rougher surface relative to the pristine chitosan beads (CH-*b*). On the contrary, the CH-*b*-Cu and CH-*b*-CL-Cu materials are characterized by heterogeneous morphologies of coarse chitosan fibrils with irregular shapes and sizes. Various factors account for the differences in the chitosan bead morphologies: (i) the mode of chitosan cross-linking, (ii) the level of Cu(II) doping onto the beads, (iii) the coordination mechanism of the Cu^2+^ ions, and (iv) the propensity of the chitosan beads to adsorb water and undergo swelling. The variable modes of interaction of Cu with the CL chitosan (CH-CL-*b*-Cu) versus the chitosan beads (CH-b-Cu, CH-*b*-CL-Cu) with and without CL are supported by FTIR and NMR spectral results, where variable morphology of the biocomposites can be inferred due to variable inter-/intramolecular interactions of the resulting complexes [53]. Previous studies [21,29] for cross-linking of chitosan reveal enhanced pillaring of the biopolymer fibrils upon cross-linking. The trend agrees with the variable bead morphology, where the CH-CL-*b*-Cu system is inferred to possess greater pillaring effects. The variable level of Cu(II) loading is supported by the dehydration process of the biocomposites, as depicted in the DTG profiles (cf. Figure 2), where the amount of bound water varies: CH-*b*-Cu > CH-*b*-CL-Cu > CH-*b* ≥ CH-CL-*b*-Cu. The thermodynamic stability of the bound water follows an opposite trend, which indicates a combination of bridge versus pendant chitosan–Cu^2+^ complexes and surface- versus pore-bound water [48].

According to the FTIR, TGA, and SEM results, it is inferred that cross-linking of chitosan before the bead formation process (hereafter referred to as in-situ cross-linking; CH-CL-*b*-Cu) may yield greater pillaring of the chitosan fibrils and reduced Cu(II) ion coordination due to competition for binding sites with the cross-linking agent, as depicted in Figure 2. A pendant model is posited due to glutaraldehyde cross-linking and reduced binding sites for Cu. In contrast, the surface-mediated cross-linked chitosan beads (CH-*b*-CL-Cu) form materials with more accessible Cu, and interstitial sites occur due to partial cross-linking of chitosan by glutaraldehyde. Both the bridge and pendant models of the coordinated Cu are possible in this biocomposite. The CH-*b*-Cu has the most accessible active sites due to pendant-coordinated Cu ions and surface-accessible chitosan sites. The rough surface morphologies for CH-*b*-Cu (without glutaraldehyde CL) and surface-cross-linked (CH-*b*-CL-Cu) chitosan beads may be due to greater loading of Cu(II) ions and partial cross-linking/grafting of glutaraldehyde onto the chitosan surface [43]. Based on the ongoing discussion, the structures of the non-cross-linked (CH-*b*), surface-(CH-*b*-CL-Cu), and in situ-(CH-CL-*b*-Cu) cross-linked chitosan beads are conceptually illustrated in Figure 2.

#### 2.1.5. X-ray Photoelectron Spectroscopy (XPS)

XPS provides useful information regarding the functional groups and their environments. The survey profiles of the XPS data (cf. Figure 5) for the CL and non-CL chitosan beads show unique spectral signatures at variable bond energies (BEs) ~285 (C1s), 400 (N1s), 532 (O1s), and 932 eV (Cu2p), with variable relative % abundances (Table 1) that concur with independent reports [54,55,56]. The C1s signal was deconvolved into three binding states assigned to the C-C/C-H (~285 eV), C-N/C-O (~286 eV), and C=N/C=O (~288 eV) from chitosan and its glutaraldehyde CL forms. According to the results in Figure 5 and Table 1, CH-*b*-Cu (47.3% and 19.4%), CH-CL-*b*-Cu (46.7% and 21.7%), CH-*b*-CL-Cu (53.8% and 18.8%) exhibit variable relative % abundances for the C-N/C-O and C=N/C=O groups, respectively. The variable trends in the relative abundances of the C-N and C=N groups for the CH-CL-*b*-Cu relate to greater CL content of this biocomposite, in agreement with the FTIR results. The increased cross-linking of CH-CL-*b*-Cu accounts for the greater abundance of the C=N group in agreement with FTIR results and Figure 2. In the case of the CH-*b*-CL-Cu biocomposite, a greater abundance of C=O groups is anticipated due to the partial cross-linking of glutaraldehyde, as described above. However, Cu(II) binding is possible at these sites, which results in diminished C=O groups, in agreement with the XPS results. The greater abundance of the C=N/C=O groups further supports greater pillaring of chitosan fibrils in the CH-CL-*b*-Cu sample, in agreement with the FTIR and SEM results.

On the other hand, the N1s spectra were resolved into two binding modes; the O=C-N (~399 eV) and C-NH_2_ (~400 eV) groups, respectively (cf. Figure 5). A Cu-N band with relative abundance of 3% (not shown) was overlapped within the O=C-N bands for the cross-linked chitosan beads that cannot be quantitatively interpreted unequivocally, due to experimental uncertainty. The greatest decrease of the C-NH_2_ groups for the CH-CL-*b*-Cu further supports the greater level of cross-linking in this material, in agreement with the above results and Figure 2. The two O1s binding energies at ~531 eV (O-Cu) and ~532 eV (-OH) provide useful information regarding the structure of the cross-linked/Cu-imbibed chitosan beads. In particular, the greater abundance of the O-Cu peak in the non-CL (CH-*b*-Cu; 24.4%) and the surface-CL (CH-b-CL-Cu; 21.3%) biocomposites relate to the greater loading of Cu ions, as supported by the FTIR Cu-O band near 600 cm^−1^. In contrast, the in-situ CL (CH-CL-b-Cu; 18.8%) had a relatively low abundance of the Cu-O groups. The latter relates to the limited functional groups in the CH-CL-*b*-Cu sample for complexation with Cu ions due to cross-linking of chitosan prior to bead formation [31]. Moreover, the greater abundance of O-H groups in the CH-CL-*b*-Cu material supports the role of the pillaring effect due to greater cross-linking, as described above (cf. Section 2.1.4), and by the FTIR and SEM results. According to the XPS and FTIR results, various factors related to the binding of Cu in the non-CL, in situ-CL, and surface-CL chitosan beads are (i) the surface-accessible functional groups (e.g., O-H, CO, and NH_2_), (ii) pore structure, and (iii) swelling capacity. The Cu-coordination in the CH-*b*-Cu, CH-*b*-CL-Cu, and CH-CL-*b*-Cu samples is depicted in Figure 3.

#### 2.1.6. Equilibrium Swelling Studies

The hydration properties of the chitosan beads were assessed using the results from solvent swelling at alkaline pH conditions (cf. Figure 6A). Swelling studies in water provide further molecular-level insight into the surface structural features and textural properties of such biomaterials, which can be inferred. The swelling capacity (%) of the chitosan beads decreased, as follows: CH-*b*-Cu (75.3%) > CH-*b*-CL-Cu (73.0%) > CH-CL-*b*-Cu (71.5%), which concurs with trends noted for the thermal stability and structure (cf. TGA, SEM, and XPS results). The water swelling results in Figure 6A concur with those described elsewhere [31]. The trend in the swelling results agrees with greater water uptake for chitosan beads that underwent bead formation prior to cross-linking with glutaraldehyde (CH-*b*-Cu and CH-*b*-CL-Cu). The foregoing indicates that water uptake by the chitosan beads is predominantly governed by the availability of dipolar functional groups of chitosan, where Cu(II) ions and interstitial domains provide secondary coordination sites. The highest Cu(II) loading for CH-*b*-Cu was previously highlighted from FTIR and XPS results and suggests that the NH_2_ and O-H functional groups are more accessible for binding with Cu(II) via a “pendant model” and a potential “bridge model” at the specified experimental conditions (cf. Figure 3a). In the case of the surface-CL chitosan beads (CH-*b*-CL-Cu), the chelation of Cu(II) follows the two models, in accordance with the pore structure and the accessibility of the NH_2_, O-H, and CO functional groups (cf. Figure 3b). By contrast, the greater pillaring of chitosan fibrils in the CH-CL-*b*-Cu sample results in the depletion of the surface-active sites, along with steric effects of the biocomposite structure that limit the adsorption of water on the bead surface and within the interstitial regions, as supported by the SEM results. The swelling studies are supported herein by XPS and TGA results, along with other studies [57,58].

### 2.2. Adsorption Studies

#### 2.2.1. Equilibrium Uptake Studies

The efficiency of the beads in the removal of phosphate ions from water was assessed through equilibrium adsorption studies. The results are presented in Figure 6B as a plot of phosphate concentration at equilibrium (C_e_) versus the amount of adsorbed phosphate (P_i_) dianions at equilibrium (Q_e_) onto the adsorbent. The results reveal a nonlinear increase for Q_e_ with increasing C_e_, where the monolayer adsorption capacity (Q_m_, mg/g) of the beads adopted the following order with P_i_: CH-*b*-Cu (133 ± 45) > CH-*b*-CL-Cu (83.9 ± 11.8) > CH-CL-*b*-Cu (80.2 ± 5.4). The equilibrium adsorption constant (K_s_, L/mg) of P_i_ for the biocomposites are listed in descending order: CH-*b*-CL-Cu (0.0298) > CH-CL-*b*-Cu (0.0267) > CH-*b*-Cu (0.0157), which agree with another study [31]. The greater uptake of P_i_ for CH-*b*-Cu and CH-*b*-CL-Cu relates to the accessibility of surface-active sites, as described above that show parallel agreement with TGA, water swelling, and SEM results. The greater value of Q_m_ for the pristine chitosan beads (CH-b-Cu) is ca. two-fold greater (133 mg/g), as compared to the P_i_ uptake for the cross-linked beads (ca. 80 mg/g). The higher uptake for the CH-*b*-Cu sample relates to the high loading of Cu(II) ions, secondary binding sites (NH, OH), and greater swelling capacity of chitosan that favors the P_i_ uptake properties of the biocomposites. In the case of the CH-CL-b-Cu, greater cross-linking was shown to attenuate the P_i_ uptake capacity, along with a decreased value of K_s_ [20,31].

#### 2.2.2. Kinetic Uptake Studies

The time-dependent adsorption-based removal of phosphate dianions from water was studied through the kinetic uptake, for which profiles are shown in Figure 6C. The kinetic results were analyzed using the pseudo-second-order (PSO) model, where the results reveal that CH-*b*-Cu displays the highest kinetic uptake capacity (21.2 ± 5.37 mg/g) of phosphate ions over a 3 h period. This trend shows parallel agreement with equilibrium adsorption studies, in accordance with the greater accessibility of surface functional groups for this biocomposite. The kinetic results also reveal that the rate of P_i_ uptake (*k_2_*, g.mg^−1^.min ^−1^) was greater for CH-*b*-CL-Cu (6.21 × 10^−4^). This observation indicates that the higher rate of P_i_ adsorption onto the surface of the cross-linked beads occurs via the accessible surface functional groups and within the pore domains, whereas the coordinated copper ions provide synergy to drive the energetics for the uptake process [37]. By contrast, the lower kinetic rates for CH-*b*-Cu and CH-CL-*b*-Cu relate to the high level of Cu loading in CH-*b*-Cu and the greater level of cross-linking in CH-CL-*b*-Cu that results in steric effects and impedes the diffusion of phosphate ions into the bead micropore domains. This trend concurs with a prolonged time required for the maximum swelling (%) of the biocomposites that were cross-linked with glutaraldehyde and Cu(II) [58]. The above trends concur with the reduced slopes observed for the isotherms for the bead systems, according to the slow rate of external and internal mass transfer, which imparts slower diffusion of the P_i_ dianions onto the beads.

#### 2.2.3. Effects of pH

Results showing the effects of pH on the sorption of phosphate anions by CH-b-Cu are shown in Figure 7 The phosphate uptake profile according to Figure 7 reveals that uptake reached a maximum (63.5 mg/g) at pH 5, 27.4 mg/g at pH 7, 36.6 mg/g at pH 9, and 37.7 mg/g at pH 11. Two factors can be used to explain the trends in the adsorption of phosphate by the chitosan beads at variable pH conditions: (1) speciation of the phosphate ions, and (2) chelation effects of Cu(II) ions. At the lowest pH conditions herein, the H_2_PO_4_^−^ anion is the main species in solution, where the adsorption affinity and capacity of Pi with the chitosan beads concurs with protonation of the amine sites and Cu^2+^ binding domains. Adsorption onto the surface and within the interstitial regions is also anticipated for these conditions. The low uptake at neutral pH concurs with the reduced protonation of chitosan since pH lies above pK_a_ (6.3-6.5) of chitosan, along with the formation of (hydr)oxides of Cu(II). At alkaline conditions the HPO_4_^2−^ and PO_4_^3−^ anions are predominant; however, the slight increase in uptake relates to the greater role of hydroxy groups on the Cu(II) centers and the biopolymer -OH/-NH_2_ groups of chitosan at these conditions.

#### 2.2.4. Regeneration Studies

The regeneration and sustainability of the biocomposite beads were tested over multiple adsorption–desorption cycles, where the results are presented in Figure 6D. The results show that CH-*b*-Cu has the highest P_i_ removal efficiency (ca. 80–85%) over four adsorption cycles, which agrees with results from equilibrium and kinetic studies. The slight increase in the removal efficiencies in cycles 3 and 4 may be related to increased swelling of the beads upon treatment with NaCl (*aq*), in agreement with salt-enhanced swelling reported for sulfated chitosan beads [59]. The results also reveal that the adsorption–desorption of P_i_ dianions from the biocomposite beads did not adversely affect the removal efficiency over the four cycles of regeneration. The use of a dilute (0.05 M) NaCl (*aq*) regenerant solution for the desorption process supports the sustainability and cost-effectiveness of these beads for phosphate remediation and recovery.

### 2.3. Comparative Studies of Other Chitosan-Based Adsorbents

Selected chitosan adsorbent systems for the removal of phosphate from water are outlined in Table 2 Among the studies presented, it is noted that the mode of preparation, the role of cross-linking on Cu(II) imbibing, and P_i_ uptake by chitosan beads are sparsely reported in the literature [31]. As well, many P_i_ uptake results for chitosan materials occur at acidic pH [17,27,60,61,62,63]. While there are reports on the modification of chitosan beads via cross-linking with glutaraldehyde and/or metal ions, the present study is unique because it provides insight into the differences in the structure and sorption properties of the chitosan beads at alkaline pH, as a consequence of the variable cross-linking strategy and incorporation of Cu(II). The study highlights the structure–adsorption properties of chitosan biocomposites cross-linked with glutaraldehyde and Cu(II) complexation.

## 3. Experimental

### 3.1. Materials

Low molecular weight chitosan (50,000–190,000 Da, ~75–85% deacetylation), sodium hydroxide (NaOH), glutaraldehyde (GH), vanadium molybdate, sodium orthophosphate, and copper sulfate were obtained from Sigma-Aldrich Canada Ltd. (Oakville, ON, Canada). Glacial acetic acid (ACS grade) was purchased from EMD Chemicals, Gibbstown, NJ, USA. All materials were used without further purification unless specified otherwise.

### 3.2. Synthesis of Copper-Imbibed Chitosan Beads

The chitosan beads (CH-*b*) were synthesized by dissolving ~2 g of chitosan in 100 mL 2% *v*/*v* glacial acetic acid with stirring. The chitosan solution was added dropwise to a 0.5 M NaOH (*aq*) via a volumetric burette (cf. Figure 1), where the resulting beads were left in NaOH (*aq*) solution for a minimum of 12 h for complete neutralization of residual acetic acid. The beads were washed with very dilute sulfuric acid solution and Millipore water for complete removal of NaOH. Copper imbibition was achieved by soaking the beads in a 0.1 M copper sulfate (*aq*) solution for a minimum of 12 h to yield the non-cross-linked beads (CH-*b*-Cu, cf. Figure 1).

In the case of the cross-linked CH Cu-imbibed beads (CH-CL-*b*-Cu, cf. Figure 1a), ~73 µL of glutaraldehyde was directly added to the CH solution with stirring for a minimum of 12 h before bead formation and Cu imbibition. On the other hand, the surface cross-linked CH beads (CH-*b*-CL-Cu, cf. Figure 1a) were prepared by soaking the prepared CH beads in 50 mL of water containing ~73 µL of glutaraldehyde for a minimum of 12 h followed by Cu(II) imbibition, as described above. All beads were dried at 60 °C for 12 h prior to characterization and further use (cf. Figure 1b). The acronyms (CH-*b*-Cu, CH-CL-*b*-Cu, and CH-*b*-CL-Cu) indicate the order of the materials modification, where the abbreviations are outlined in the caption for Figure 1.

### 3.3. Characterization

#### 3.3.1. Thermogravimetric Analysis (TGA)

The thermal stability of the beads was determined by employing a TA Instruments Q50 TGA (New Castle, DE, USA). The analysis conditions are listed as heating rate: 5 °C min^−1^ to a maximum temperature of 500 °C, and a nitrogen purge gas. The results were presented as a DTG plot of weight with temperature (%/°C) versus temperature (°C), as well as a TGA plot of mass % vs. temperature.

#### 3.3.2. Fourier Transform Infrared (FTIR) Spectroscopy

A Bio-RAD FTS-40 IR spectrophotometer (Santa Clara, CA, USA) was used for the acquisition of the FTIR spectra of the beads. Prior to spectra acquisition, the crushed beads were mixed with pure spectroscopic-grade KBr in a weight ratio of 1:10. The spectra were obtained in reflectance and DRIFT (diffuse reflectance infrared Fourier transform) modes at 295 K with a resolution of 4 cm^−1^. A total of 100 scans were recorded over the 400–4000 cm^−1^ spectral range and corrected relative to a background of pure KBr.

#### 3.3.3. Scanning Electron Microscopy (SEM)

The SEM images of the beads were acquired using a scanning electron microscope (SEM; Model SU8000, HI-0867-0003, Tokyo, Japan). The beads were coated with a 5 nm layer of chromium prior to image acquisition according to the following instrument conditions: accelerating voltage (3 kV), working distance (6.1–6.7 mm), and magnifications of 150× and 25,000×.

#### 3.3.4. Solid State ^13^C NMR Spectroscopy

^13^C solids NMR spectra of the samples were collected using a Bruker AVANCE III HD spectrometer (Billerica, MA, USA) equipped with a 4 mm DOTY CP-MAS (cross polarization with magic angle spinning) solid probe operating at 125.77 MHz (^1^H spectral frequency at 500.23 MHz). A spinning speed of 7.5 kHz, a ^1^H 90° pulse of 3.5 µs, and a contact time of 1 ms with a ramp pulse on the ^1^H channel were used for the acquisition of the ^13^C solids NMR spectra. A total of 6000–28,000 scans were accumulated with a recycle delay of 2 s, depending on the sample. All experiments were recorded using 71 kHz SPINAL-64 decoupling during acquisition with external reference to adamantane at 38.48 ppm (low field signal).

#### 3.3.5. X-ray Photoelectron Spectroscopy (XPS)

A Kratos (Manchester, UK) AXIS Supra system equipped with a 500 mm Rowland circle monochromated Al K-α (1486.6 eV) source, combined with a hemispherical analyzer (HSA) and a spherical mirror analyzer (SMA) was used to acquire the XPS profiles of the beads. All survey scan spectra were collected in the −5–1200 binding energy range in 1 eV steps with a pass energy of 160 eV, using a spot size of a hybrid slot (300 × 700 microns). For the high-resolution analysis, scans of several regions were also conducted using 0.05 eV steps and a pass energy of 20 eV. An accelerating voltage of 15 keV and an emission current of 15 mA were used for the analysis.

#### 3.3.6. Equilibrium Swelling Studies

The equilibrium solvent swelling properties of the beads were determined by equilibrating the beads (~50 mg) in 30 mL of Millipore water in a horizontal shaker for ~48 h. The weights of the hydrated beads (w_s_) were determined after the removal of water from the surface of the beads by dabbing with a dry filter paper. The beads were dried in an oven at 40 °C to a constant weight (w_d_; ±0.01 g). The percentage swelling of the beads was calculated according to Equation (1):(1)$$Sw\left(\%\right)=\frac{w_{s}-W_{d}}{W_{d}}\times 100$$

### 3.4. Adsorption Studies

#### 3.4.1. Equilibrium Sorption of Phosphate Anions

A 500 ppm orthophosphate (P_i_) solution containing 353 ppm hydrogen phosphate anion species was prepared by dissolving an appropriate mass of sodium orthophosphate in water with stirring. The pH of the resulting solution was adjusted to 8.5 using 0.1 M NaOH (*aq*). A series of 10 mL phosphate solutions (14–500 ppm) was prepared in 10-dram vials, where a fixed dosage (10 mg) of the beads was added to each vial. The sample vials were equilibrated in a horizontal shaker at 295 K for 24 h, where the mixing rate was ~172 rpm. The P_i_ removal capacity of the beads (Q_e_) was calculated according to Equation (2) by determining the initial (C_o_) and final (C_e_) concentrations of P_i_. Vanadate molybdate reagent was used as the color-developing reagent while a double-beam spectrophotometer (Varian CARY 100) was used to determine the concentration of the phosphate solutions at 295 ± 0.5 K.
(2)$$Q_{e}=\frac{\left(C_{o}-C_{e}\right)\times V}{m}$$

#### 3.4.2. Kinetic Uptake Studies

Kinetic studies were carried out using a one-pot method [68] as follows: ~40 mg of the beads was added to 120 mL of a 71 ppm P_i_ solution. At designated intervals, 3 mL aliquots of the P_i_ solution were sampled and the residual concentration of P_i_ in the aliquots was determined at 295 ± 0.5 K, using a double-beam spectrophotometer (Varian CARY 100). Uptake of P_i_ anions in the aliquots at each sampling time interval (t) was estimated according to Equation (3), where C_o_ and C_t_ refer to the concentration of P_i_ at t = 0 and variable time (t).
(3)$$Q_{t}=\frac{\left(C_{o}-C_{t}\right)\times V}{m}$$

### 3.5. Regeneration Studies

Regeneration of the beads was carried out using 10 mL of 0.05 M NaCl (*aq*) as the regenerant. The adsorption–desorption cycle was repeated four times, and the removal efficiency of the regenerated beads was determined, as described in Section 3.4.1.

## 4. Conclusions

Chitosan biocomposite beads (CH-*b*-Cu, CH-CL-*b*-Cu, and CH-*b*-CL-Cu) were modified by two modes of cross-linking with glutaraldehyde along with Cu(II) complex formation (cf. Figure 1). The biocomposites were characterized by TGA and spectroscopy (FTIR and ^13^C solids NMR), which reveal variable cross-linking between glutaraldehyde and the chitosan NH_2_ groups, along with Cu(II) ion complex formation within the bead microstructure; this includes spectral shifts for key IR signatures, variation in thermal gravimetry profiles, and broadening of ^13^C solids NMR resonance lines. SEM images provide support that the biocomposite has different modes of cross-linking that yield distinct bead morphologies. Notably, the CH-CL-*b*-Cu system showed enhanced pillaring of chitosan fibrils, aligning with higher levels of glutaraldehyde cross-linking. XPS results further confirmed that the physicochemical properties of beads vary with the cross-linking mode, where beads with greater cross-linking exhibited lower Cu(II) loading. The equilibrium and kinetic results for P_i_ adsorption, alongside solvent swelling studies, indicated that the cross-linking modality significantly influenced the beads’ physicochemical properties. The latter was evidenced by the variation in the P_i_ adsorption capacity of the biocomposites. Overall, this study contributes to the development of cost-effective and sustainable biocomposite adsorbents for equilibrium and dynamic separations, which demonstrate their efficacy at alkaline pH for the removal of phosphate and related oxyanions from aqueous media.

## Data Availability

Data will be made available upon reasonable request.

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
