# Peer review of "Chitosan Biocomposites with Variable Cross-Linking and Copper-Doping for Enhanced Phosphate Removal"

_molecules, 2024, doi:10.3390/molecules29020445_

Round 1

Reviewer 1 Report

Comments and Suggestions for Authors

Authors have attempted to use chitosan based bicomposites for phosphate remediation with and without cross linkers. Material is well characterized and presented. The cross linking doesnot seem to have any positive effect on the performance. Henceforth the findings were not much of interest to the  said application. Reserch content is fair and good quality. Below comments needs to be addressed.

Typo Errors

Figure 2 is given incorrect caption. These are TG-DTA plots. Caption to be correvted. Mass% vs Temp should also be given as separate plot.

Scheme 2 - Chitosan or XX - missing part in the name of component

Queries

Leaching of copper from the adsorbent was tested? What if the sorbent loses copper in usage? - Results should be presented.

Why copper sulphate was choosen as the candidate? Any further treatment to remove the sulphates? If not what happened to the sulphates from the precursor?

What form of copper was present in the adsorbent? 

TGA was done till 500C - plots should also be extented to the same temperature

Chitosan doesnt withstand temperature till 500C. Another decompostion is expected - this peak is missing in the TGA results.

Performance with Change in pH can also be explored and presented.

Synthesis water remediation is easy. Authors are encouraged to work with the real life waste water - primarily the effect of co-existing ions will deeply affect the performance of adsorbent.

Spent adsorbent should be tested for any chages in the properties 

Overall the work is good. Minor changes as per the comments to be amended to get the manuscript in publishable form.

Author Response

Authors’ Response to Reviewer Comments on Manuscript ID: molecules-2767555

The authors’ response to the queries are outlined in red font in a point-by-point fashion for the various reviewer comments & queries for each reviewer, as outlined below.

Reviewer 1

Authors have attempted to use chitosan based bicomposites for phosphate remediation with and without cross linkers. Material is well characterized and presented. The cross linking doesnot seem to have any positive effect on the performance. Henceforth the findings were not much of interest to the said application. Reserch content is fair and good quality. Below comments needs to be addressed.

Typo Errors

Figure 2 is given incorrect caption. These are TG-DTA plots. Caption to be correvted. Mass% vs Temp should also be given as separate plot.

Response: We thank the reviewer for this comment. The caption for figure 2 has been corrected. The mass% vs Temp plot was not presented in the original manuscript; however, the mass% data is added to the revised manuscript.

Scheme 2 - Chitosan or XX - missing part in the name of component

Response: We thank the reviewer for this comment. The naming in the scheme has been revised to address the query.

Queries

Leaching of copper from the adsorbent was tested? What if the sorbent loses copper in usage? - Results should be presented.

Response: We thank the reviewer for this comment. The focus of the study was to evaluate the effects of various modes of cross-linking on copper doping and adsorption properties of the beads. Even though the Cu-doped materials show good structural integrity (based on SEM, FT-IR, TGA, etc.), the authors agree that copper leaching could potentially occur (especially during the adsorption and kinetics studies). However, the scope of the current study covers the design of the materials and the characterization of their structure-property relationship. A study of the material structural integrity with respect to copper leaching in different conditions of agitation, temperature, pH, etc. is planned as a follow-up study.

Why was copper sulphate chosen as the candidate? Any further treatment to remove the sulphates? If not what happened to the sulphates from the precursor?

Response: Copper sulphate was chosen due to the efficient binding of sulphate groups to the functional groups of chitosan. Several studies have demonstrated the crosslinking of chitosan with sulphate groups. The authors anticipated that the crosslinking of chitosan with sulphate groups will enhance the stability of the beads.

What form of copper was present in the adsorbent? 

Response: XPS results supports the existence of copper in the +2-oxidation state. This information has been added/updated in the revised manuscript

TGA was done till 500C - plots should also be extended to the same temperature.

Response: The TGA was done till 500 °C to make sure no information was missed. However, since the materials decompose at T < 400 °C, there is no useful welght loss events above 400 °C, which explains why that region was not presented in the plots.

Chitosan doesn’t withstand temperature till 500C. Another decomposition is expected - this peak is missing in the TGA results.

Response: Studies have shown that chitosan decomposition does not extend beyond 400 °C, however there may be peaks above 400 °C due to copper (doi:10.1002/app.38247). The interest of the authors was on the effects of crosslinking and copper binding onto the chitosan domains of the composites. Hence, the focus of the TGA profile was placed on the 0 – 400 °C region.

Performance with Change in pH can also be explored and presented.

Response: We thank the reviewer for this comment. The authors agree that pH effects on the adsorption properties of the composites is important and will consider it during the second phase of the studies that will focus on optimization of adsorption conditions, as mentioned previously.

Synthesis water remediation is easy. Authors are encouraged to work with the real life waste water - primarily the effect of co-existing ions will deeply affect the performance of adsorbent.

Response: We thank the reviewer for the thoughtful comment. The authors are considering this area, along with looking at the structural integrity of the materials with respect to Cu leaching, in future follow-up work, which extends beyond the objectives of the current study..

Spent adsorbent should be tested for any changes in the properties 

Response: The authors are grateful to the reviewer for this comment. This area will be included in the follow-up study.

Overall the work is good. Minor changes as per the comments to be amended to get the manuscript in publishable form.

Summary response: The authors appreciate the insightful and constructive comments provided by reviewer #1, along with the opportunity to improve this manuscript submission. The manuscript was further edited for language, syntax, and clarity to meet the high standards of this journal.

Reviewer 2 Report

Comments and Suggestions for Authors

Authors must address the following comments before being considered for publication:

1. The introduction section should be organized to highlight the novelty of your work. There are many reports on the functionalization of chitosan and preparation of cross-linkers for composite materials in the removal of metal ions. See this reference:
https://doi.org/10.1007/s11356-023-29896-6
2. The authors should improve scheme 2.
3. Should the authors explain why the removal efficiencies in cycles 3 and 4 increase?
4. The authors did not use the appropriate reference format for this journal.
5. The quality of the SEM images Figure 2 is poor, they should indicate scale.
6. Authors must show EDX studies
7. The authors should explain the effect of pH on the removal efficiency.

Author Response

Authors’ Response to Reviewer Comments on Manuscript ID: molecules-2767555

The authors’ response to the queries are outlined in red font in a point-by-point fashion for the various reviewer comments & queries for each reviewer, as outlined below.

Reviewer 2

Comments and Suggestions for Authors

Authors must address the following comments before being considered for publication:

1. The introduction section should be organized to highlight the novelty of your work. There are many reports on the functionalization of chitosan and preparation of cross-linkers for composite materials in the removal of metal ions. See this reference:
https://doi.org/10.1007/s11356-023-29896-6

Response: We appreciate the reviewer for this comment. The introduction of the paper has been revised to highlight the novelty of the work according to the reviewer’s recommendation.

  1. The authors should improve scheme 2.

Response: We thank the reviewer for this comment. Scheme 2 has been improved in the revised manuscript, as recommended.

  1. Should the authors explain why the removal efficiencies in cycles 3 and 4 increase?

Response: The slight increase in the removal efficiencies in cycles 3 and 4 may be related to enhanced swelling of the beads due to the treatment with NaCl. Shu & Zhu have reported on the swelling of sulphated chitosan beads due to treatment with NaCl (see the following reference: https://www.sciencedirect.com/science/article/pii/S0378517301009437).

  1. The authors did not use the appropriate reference format for this journal.

Response: The references have been updated in accordance with the journal formatting requirements in the revised manuscript,

  1. The quality of the SEM images Figure 2 is poor; they should indicate scale.

Response: Scaling is now included in the SEM images of the revised manuscript, along with insets at variable magnification.

  1. Authors must show EDX studies

Response: We thank the reviewer for this comment. The characterization of the materials in the current study is addressed using FT-IR, NMR, and XPS in accordance with the objectives of this study. The XPS results are considered to provide valuable quantitative analysis of the various atomic elements, in a similar fashion as EDX results.

  1. The authors should explain the effect of pH on the removal efficiency.

Response: We thank the reviewer for this comment. The authors agree that pH effects on the adsorption properties of the composites is important. The current study was focused on pH 8.5 to focus on the adsorption of phosphate dianion species. Additional pH conditions will be considered in follow-up studies that will focus on optimization of adsorption conditions across a wider range of variables, as mentioned in the responses for Reviewer #2.

Reviewer 3 Report

Comments and Suggestions for Authors

1.     For the adsorption of phosphorus, the common way at present is the adsorption method. Because of its low cost and wide source of raw material for adsorption. Therefore, here are some adsorption work for phosphorus from our group, which is of good reference for the author's article. It is recommended to cite some or all of them. They are as follows:

(1)   Magnetic aminated lignin/CeO2/Fe3O4 composites with tailored interfacial chemistry and affinity for selective phosphate removal, Xiang-Cheng Shan, Yu-Meng Zhao, Shu-Feng Bo, Li-Yu Yang, Qing-Da An, Zuo-Yi Xiao, Shang-Ru Zhai* Total Environmental, 2021, 796, 148984.

(2)   Hollow polyethyleneimine/carboxymethyl cellulose beads with abundant and accessible sorption sites for ultra-efficient chromium (VI) and phosphate removal, H.R. Yang, S.S. Li, S.S. Chen, Y.S. An, H.R. Yang YANG Huarong, LI Shanshan, SIN Xiangcheng, CHEN Yang, AN Qingda, XIAO Zuoyi, ZHAI Shangru Separation and Purification Technology, 2022, 278, 119607.

2.     Pictures are an important means of reflecting data, so the standardization and rigor of pictures are important for improving the grade of the whole article. Here are some graphical suggestions:

1) The primary and secondary scales of the axes should be the same for all images.

2) Option 2 is partially incomplete. This greatly affects the presentation of the experimental scheme. In addition, there should be a consistent specification for some of the chemical bonds as well as the functional groups plotted.

3.     Changing the order of cross-linking is an important means of verifying the adsorption properties of a material, especially for the preparation of gel microspheres. In addition, have the authors considered determining the next best adsorption property by increasing or decreasing a variable or by changing the amount of raw material? Incidentally, the synergistic effect between the materials constituting the adsorbent was verified.

4.     Ion coexistence experiments are important to verify the adsorption potential of adsorbents in complex waters. Anions similar to phosphate mixed to form a binary adsorption system, which has an extremely strong adsorption capacity.

5.     Phosphate exists in different forms at different pH values. Therefore, pH related inquiries are essential. It also helps in the interpretation of the experimental mechanism.

6.     It is recommended to perform relevant thermodynamic experiments at several different temperature conditions and calculate ∆G, ∆H, and ∆S for an in-depth explanation of the mechanism.

7.     Figure 7c, Adsorption kinetics shows the relevant data for adsorption kinetics. More importantly, the data need to be further processed to complete the fitting of T versus lg(Qe-Qt) and T versus t/Qt curves to elucidate the kinetic experiments more intuitively.

8.      Recommendations 4-8 are relevant validation experiments that need to be added and presented in the text. Specifically, the first recommendation can be referred to.

Author Response

Authors’ Response to Reviewer Comments on Manuscript ID: molecules-2767555

The authors’ response to the queries are outlined in red font in a point-by-point fashion for the various reviewer comments & queries for each reviewer, as outlined below.

Reviewer 3

Comments and Suggestions for Authors

  1. For the adsorption of phosphorus, the common way at present is the adsorption method. Because of its low cost and wide source of raw material for adsorption. Therefore, here are some adsorption work for phosphorus from our group, which is of good reference for the author's article. It is recommended to cite some or all of them. They are as follows:

(1)   Magnetic aminated lignin/CeO2/Fe3O4 composites with tailored interfacial chemistry and affinity for selective phosphate removal, Xiang-Cheng Shan, Yu-Meng Zhao, Shu-Feng Bo, Li-Yu Yang, Qing-Da An, Zuo-Yi Xiao, Shang-Ru Zhai* Total Environmental, 2021, 796, 148984.

(2)   Hollow polyethyleneimine/carboxymethyl cellulose beads with abundant and accessible sorption sites for ultra-efficient chromium (VI) and phosphate removal, H.R. Yang, S.S. Li, S.S. Chen, Y.S. An, H.R. Yang YANG Huarong, LI Shanshan, SIN Xiangcheng, CHEN Yang, AN Qingda, XIAO Zuoyi, ZHAI Shangru Separation and Purification Technology, 2022, 278, 119607.

Response: We thank the reviewer for the suggestion. These citations are included in the revised manuscript.

  1. Pictures are an important means of reflecting data, so the standardization and rigor of pictures are important for improving the grade of the whole article. Here are some graphical suggestions:

1) The primary and secondary scales of the axes should be the same for all images.

2) Option 2 is partially incomplete. This greatly affects the presentation of the experimental scheme. In addition, there should be a consistent specification for some of the chemical bonds as well as the functional groups plotted.

Response: We thank the reviewer for this constructive comment. The figures are now revised.

  1. Changing the order of cross-linking is an important means of verifying the adsorption properties of a material, especially for the preparation of gel microspheres. In addition, have the authors considered determining the next best adsorption property by increasing or decreasing a variable or by changing the amount of raw material? Incidentally, the synergistic effect between the materials constituting the adsorbent was verified. 

        Response: We thank the reviewer for this constructive comment. This aspect of work is outside the scope of the current work and will be considered in a follow-up study for this project.

  1. Ion coexistence experiments are important to verify the adsorption potential of adsorbents in complex waters. Anions similar to phosphate mixed to form a binary adsorption system, which has an extremely strong adsorption capacity.

Response: We thank the reviewer for this comment. This aspect of work is outside the scope of the current work and will be considered in the next phase of the project which will deal with optimizing the adsorption properties and the corresponding synthetic conditions.

  1. Phosphate exists in different forms at different pH values. Therefore, pH related inquiries are essential. It also helps in the interpretation of the experimental mechanism.

Response: We thank the reviewer for this comment. The authors agree that pH effects on the adsorption properties of the composites is important and will consider it during future studies that will be focused on optimization of the adsorption conditions.

  1. It is recommended to perform relevant thermodynamic experiments at several different temperature conditions and calculate ∆G, ∆H, and ∆S for an in-depth explanation of the mechanism.

 Response: We thank the reviewer for this comment. This aspect of work is outside the scope of the current study and will be considered in the next phase of the project which will deal with optimizing the adsorption and the synthetic conditions, along with an evaluation of the adsorption mechanism.

  1. Figure 7c, Adsorption kinetics shows the relevant data for adsorption kinetics. More importantly, the data need to be further processed to complete the fitting of T versus lg(Qe-Qt) and T versus t/Qt curves to elucidate the kinetic experiments more intuitively.

Response: We thank the reviewer for this comment. At the stage of the current work, the authors were mainly focused on elucidating how the various mode of crosslinking and level of copper doping affects the adsorption rate which was duly obtained through the plot of qt vs time. The authors will consider the suggestion in future work, which deals with optimizing the adsorption and synthetic conditions.

  1. Recommendations 4-8 are relevant validation experiments that need to be added and presented in the text. Specifically, the first recommendation can be referred to.

        Response: We thank the reviewer for the comment. The authors agree that the recommended experiments will add value to the work. However, these experiments are outside the scope of the current study which focused on elucidation of the effects of variable crosslinking strategies via copper complexation and adsorption properties of chitosan beads. The suggested experiments will be considered as part of future studies.

Summary: The authors appreciate the insightful and constructive comments provided by reviewer #3, along with the opportunity to improve this manuscript submission. The manuscript was further edited for language, syntax, and clarity to meet the high standards of this journal.

Reviewer 4 Report

Comments and Suggestions for Authors

The manuscript describes chitosan based beads that were prepared by different crosslinking experiments (covalent and copper complexation) followed by the use of these materials for the absorption of phosphate. Unfortunately, the actual purpose and direction of this work is not clear. Similar approaches for crosslinking / bead formation of chitosan have been reported before. This is not a fundamental work that is trying to tailor properties, e.g., by systematically studying the effect of different amounts of copper or crosslinker. It also does not elucidate the molecular structures that are formed. The authors speculate on the nature and chemical structure of Cu-complexation using FTIR, solid-state NMR and SEM experiments. The conclusions are not convincing and mere speculations. The spectra all look very similar and the resolution is not very high. The SEM images don’t enable comparison because scale bars / magnification factors are missing / not readable. The only actual contribution in this context come from the XPS experiments. The final materials show slightly different adsorption behavior but with little convincing explanation how these occurred. Moreover, some of the data require more statistical evaluation. Sustainability and applicability of the materials is questionable considering the use of larger amounts of copper. Studies to demonstrate that not leaking occurs are missing.

Finally, the manuscript has technical flaws, such as incorrect figure captions and flawed molecular structures. Overall, this manuscript is mediocre. I see little contribution to the journal scope. Another journal with an engineering focused scope would be more suited. I recommit to reject the manuscript due to these reasons.

The following major issues should be considered:

- L 11: This is not a copolymer.

- L 13: SEM is not a spectroscopy method.

- L 19: Whether a material is sustainable or not depends on its (bio)origin and environmental fait and not if it can be reused to a certain extent.

- L 26: The materials contain a lot of heavy metal ions. Unless it is demonstrated that NO copper is leaking, application in environmental technologies is questionable.

- L 52: The natural polysaccharide chitin is abundant, not the man-made derivative chitosan. Moreover, chitosan is NOT a (permanently) charged polymer. It might be partly charged, when in aqueous media, depending on the pH value.

It should be “amino groups” not “amine groups”. Moreover, the authors should not switch between amine / amino groups and NH2 groups.

- L 53: The correct IUPAC term is “hydroxy” groups (without “l”). Moreover, no switching between hydroxy and OH groups.

- L 72: The motivation for introducing copper ions is not clear and has to be explained. Apparently, the authors want to increase cross-linking but why is this desired? Copper complexation could have an adverse effect on the phosphate adsorption because it “blocks” amino groups. The authors should comment on this here.

- L 84: I see no modularity here. The authors follow a straight step-by-step approach. I see no possibility to branch out and obtain different (modular) materials.

- L 90: “Low” is too subjective. What is the actual molecular weight?

- L 101: An actual concentration is needed. Moreover, I a missing a discussion on how much of the copper ions applied are actually contained in the final material (an how much has to be disposed as heavy metal waste).

- L 198: I do not understand this heading. Chitosan contains nitrogen and so do ALL chemically modified chitosan derivatives. Furthermore, the compound reported herein are no chemical derivatives because the authors prepared insoluble / crosslinked materials.

- L 254: Again, this is not a copolymer.

- L 276: The figure caption is incorrect.

- L 277: The solid state NMR spectra are not very informative an don’t enable such detailed conclusions that where drawn here.

- L 293: The drawing of the molecular structure is incorrect / not appropriate for publication. The C2-C3 bond has to be a bold / thick bond because it is in the front and not pointing out of the plane as depicted here by the arrow like bond. In the right repeating unit, the bonds C1-O, O-C5, and C5-C4 are in the front and have to be depicted bold. The “n” in the bracket is incorrect because it implies that (a) the number of repeating units / DP has to be an even number and (b) the drawing implies that chitosan has a disaccharide repeating unit with alternating acetylated and non-acetylated amino groups. The authors should just depict ONE repeating unit with a -NHR group and define R as either H or acetyl.

- L 294: The quality / resolution of the SEM is not very good. It is difficult to get detailed information, especially because scale bares and magnification are missing.

- L 300: I cannot see any linear fibrils in the images. Without any scale bars it is difficult to get a perspective.

- L 302: The comments on the different modes of crosslinking are very vague and speculative. I don’t see how these conclusions can be drawn from the presented SEM images.

- L 314: I cant see how conclusions on any structural features ( such as bridged vs. pendant complexes) can be drawn from these SEM images. This is mere speculation.

- L 319: This scheme is not acceptable for publication. Parts of the text (“Chitosan ???”) are cut off. One of the nitrogen atoms in the chitosan structure features an -CH=O group, which I suppose should be an acetyl group. The atom labels features a broad variety of font sizes, which makes the image confusing.

- L 324 to 335: This is again, broad speculation. It would be very difficult to draw these elaborate conclusion on the molecular structures within the crosslinked materials even if the SEM images had much higher quality. The spectroscopic IR and NMR data are not very helpful either.

- L 378: The scheme has similar flaws and errors as described above. Fond sizes should be unified. The copper ion is missing its 2+ charge and some copper ions feature unspecified bonds (probably covalent). The bond between copper 2+ and water is a solid line, which implies a covalent rather than a complex bond. Structure c has some blue lines scribbled into the molecular structure. These look like covalent bonds with a long alkyl chain. Apparently, these should symbolize steric hindrance, which is very confusing because the same color and thickness is used as for the covalent bonds. Moreover, it is not clear from this picture where this supposed steric hindrance comes from.

- L 384: The water uptake values are practically identical considering the large error bars. Any conclusions on the chemical structures within the beads based on these values are void and mere speculation, unless the authors can demonstrate significant differences and a clear trend using statistical analysis.

- L 410: Correlation with water uptake values are void (see comment above).

- L 435: I see no experimental proof for greater accessibility here (or in the SEM images).

- L 450: The presentation of these data is confusing and not uniform (e.g., different font sizes, non-uniform axis titles). The corresponding discussion of the data is questionable.

The graphs in figure 6A are very similar considering the error bar. A statistical test (which is missing) would probably reveal no statistical relevance, i.e., the swelling of all bead types is practically identical. The same can be assume for the date in 6D. The values for all beads are almost identical in most cases. Unfortunately, error bars are missing. Moreover, it is not clear here what “% removal” means (same goes for Qe and Qt. Figures (and table) need to be comprehendible without reference to the text.

What mathematical model(s) were use to fit curves to the scattered data points?

The SI unit for “minutes” is “min” (not “mins”).

- L 455: As described above. Error bars and statistical analysis are needed. The values are very close together. Why does the removal efficiency increase in cycles 3 and 4? The differences are larger than the supposed differences between the different bead types. Moreover, I am not convinced that the whole process is sustainable considering that a lot of Cu2+ is used. How about leakage of cupper ions over time / the different cycles?

- L 494: I am not fully convinced that these absorbents are sustainable. The preparation requires immersion in an aqueous copper solution that needs to be disposed as heavy metal waste afterwards. Furthermore, the authors have not studied leaking of copper in the different cycles, which is basic requirement before application can be considered.

- L 497: This claim is not justified and I cannot understand how the authors came to this conclusion. They studied ONE pH value. By comparison, most of the cited references reported a broader range of pH values.

Author Response

Authors’ Response to Reviewer Comments on Manuscript ID: molecules-2767555

The authors’ response to the queries are outlined in red font in a point-by-point fashion for the various reviewer comments & queries for each reviewer, as outlined below.

Reviewer 4

Comments and Suggestions for Authors

The manuscript describes chitosan-based beads that were prepared by different crosslinking experiments (covalent and copper complexation) followed by the use of these materials for the absorption of phosphate. Unfortunately, the actual purpose and direction of this work is not clear. Similar approaches for crosslinking / bead formation of chitosan have been reported before. This is not a fundamental work that is trying to tailor properties, e.g., by systematically studying the effect of different amounts of copper or crosslinker. It also does not elucidate the molecular structures that are formed. The authors speculate on the nature and chemical structure of Cu-complexation using FTIR, solid-state NMR and SEM experiments. The conclusions are not convincing and mere speculations. The spectra all look very similar, and the resolution is not very high. The SEM images don’t enable comparison because scale bars / magnification factors are missing / not readable. The only actual contribution in this context come from the XPS experiments. The final materials show slightly different adsorption behavior but with little convincing explanation how these occurred. Moreover, some of the data require more statistical evaluation. Sustainability and applicability of the materials is questionable considering the use of larger amounts of copper. Studies demonstrate that non-leaking occurs are missing.

Finally, the manuscript has technical flaws, such as incorrect figure captions and flawed molecular structures. Overall, this manuscript is mediocre. I see little contribution to the journal scope. Another journal with an engineering focused scope would be more suited. I recommit to reject the manuscript due to these reasons.

Response: We thank the reviewer for the constructive comments and recommendations. The technical flaws have been corrected in the revised manuscript. However, the authors believe that the results of the experiments presented in the study provide sufficient support to support the conclusions related to elucidation of the effects of variable crosslinking strategy on copper complexation and the adsorption properties of chitosan beads at alkaline pH conditions.

The following major issues should be considered:

- L 11: This is not a copolymer.

Response: We thank the reviewer for this comment. The authors agree with the reviewer and the word copolymer has been removed from the revised manuscript

- L 13: SEM is not a spectroscopy method.

 Response: We agree with the reviewer. SEM has been removed from spectroscopic techniques in the revised manuscript.

- L 19: Whether a material is sustainable or not depends on its (bio)origin and environmental fait and not if it can be reused to a certain extent.

Response: We thank the reviewer for this comment. However, the authors believe that material sustainability relates to its origin, environmental fate, and reusability. Chitosan, which is the major component of the materials reported in this study, is a biopolymer derived from chitin.

- L 26: The materials contain a lot of heavy metal ions. Unless it is demonstrated that NO copper is leaking, application in environmental technologies is questionable.

Response: We thank the reviewer for this comment and agree that copper leaching can be a major source of environmental concern. The authors have noted this suggestion and will incorporate it during the follow-up work on the techno-economic feasibility of the biocomposite adsorbents which will focus on the optimization of the synthetic process through adsorption and kinetic studies at variable T, pH, conditions, including using real environmental water samples.

- L 52: The natural polysaccharide chitin is abundant, not the man-made derivative chitosan. Moreover, chitosan is NOT a (permanently) charged polymer. It might be partly charged, when in aqueous media, depending on the pH value.

It should be “amino groups” not “amine groups”. Moreover, the authors should not switch between amine / amino groups and NH2 groups.

Response: The reviewer’s comment is noted. While some of the references were an oversight/typos from the authors’ end, the term ‘NH2 groups’ is now used throughout the revised manuscript. The authors agree with the reviewer than chitin, which is the source of chitosan, is the most abundant polysaccharide after cellulose.

- L 53: The correct IUPAC term is “hydroxy” groups (without “l”). Moreover, no switching between hydroxy and OH groups.

Response: We thank the reviewer for this comment. Hydroxyl has been replaced with hydroxy in the revised manuscript.

 - L 72: The motivation for introducing copper ions is not clear and has to be explained. Apparently, the authors want to increase cross-linking but why is this desired? Copper complexation could have an adverse effect on the phosphate adsorption because it “blocks” amino groups. The authors should comment on this here.

Response: We thank the reviewer for this comment. The motivation for the use of copper in now included in the revised manuscript.

- L 84: I see no modularity here. The authors follow a straight step-by-step approach. I see no possibility to branch out and obtain different (modular) materials.

Response: We thank the reviewer for this comment. The modularity in the approach applies to the variable nature of cross-linking strategy employed in the study.

- L 90: “Low” is too subjective. What is the actual molecular weight?

 Response: The molecular weight of chitosan is now included in the revised manuscript.

- L 101: An actual concentration is needed. Moreover, I a missing a discussion on how much of the copper ions applied are actually contained in the final material (an how much has to be disposed as heavy metal waste).

Response: We thank the reviewer for this comment. 0.1 M Cu solution was used for the experiments. XPS results show that the different beads contain different concentrations of copper ions. The goal of the study was to elucidate if the different cross-linking strategies will result in variable copper doping of the beads and XPS results affirmed this. Furthers studies will be performed to ascertain the actual concentration of copper ions in the beads during the next phase of the project which will focus on optimization of the synthetic and adsorption parameters.

- L 198: I do not understand this heading. Chitosan contains nitrogen and so do ALL chemically modified chitosan derivatives. Furthermore, the compound reported herein are no chemical derivatives because the authors prepared insoluble / crosslinked materials.

Response: We thank the reviewer for this comment. The heading was added in error and has been removed in the revised manuscript.

- L 254: Again, this is not a copolymer.

Response: The word copolymer has been removed from the entire manuscript.

- L 276: The figure caption is incorrect.

 - L 277: The solid state NMR spectra are not very informative an don’t enable such detailed conclusions that where drawn here.

 Response: The caption was corrected in the revised manuscript, along with edits to the discussion and conclusion sections that are supported by the solids NMR results presented in this study.

- L 293: The drawing of the molecular structure is incorrect / not appropriate for publication. The C2-C3 bond has to be a bold / thick bond because it is in the front and not pointing out of the plane as depicted here by the arrow like bond. In the right repeating unit, the bonds C1-O, O-C5, and C5-C4 are in the front and have to be depicted bold. The “n” in the bracket is incorrect because it implies that (a) the number of repeating units / DP has to be an even number and (b) the drawing implies that chitosan has a disaccharide repeating unit with alternating acetylated and non-acetylated amino groups. The authors should just depict ONE repeating unit with a -NHR group and define R as either H or acetyl.

 Response: The drawing of the molecular structure has been revised as recommended.

- L 294: The quality / resolution of the SEM is not very good. It is difficult to get detailed information, especially because scale bares and magnification are missing.

 Response: Scale bars are now included in the revised figures.

- L 300: I cannot see any linear fibrils in the images. Without any scale bars it is difficult to get a perspective.

 Response: Scale bars are now included in the revised figures.

- L 302: The comments on the different modes of crosslinking are very vague and speculative. I don’t see how these conclusions can be drawn from the presented SEM images.

 Response: Scale bars are now included in the SEM images and insets at higher magnification, which provide support for the conclusions of this study.

- L 314: I cant see how conclusions on any structural features (such as bridged vs. pendant complexes) can be drawn from these SEM images. This is mere speculation.

Response: We thank the reviewer for this comment. However, the conclusions were not drawn from SEM images, but from FTIR and NMR results while the differences in the morphology of the materials supports the FTIR and NMR Results. The SEM discussions were revised accordingly.

- L 319: This scheme is not acceptable for publication. Parts of the text (“Chitosan ???”) are cut off. One of the nitrogen atoms in the chitosan structure features an -CH=O group, which I suppose should be an acetyl group. The atom labels features a broad variety of font sizes, which makes the image confusing.

Response: We thank the reviewer for the comment. The structure was revised accordingly.

- L 324 to 335: This is again, broad speculation. It would be very difficult to draw these elaborate conclusion on the molecular structures within the crosslinked materials even if the SEM images had much higher quality. The spectroscopic IR and NMR data are not very helpful either.

Response: The discussion and conclusions have been updated to reflect the support provided by the spectral results presented herein.

- L 378: The scheme has similar flaws and errors as described above. Fond sizes should be unified. The copper ion is missing its 2+ charge and some copper ions feature unspecified bonds (probably covalent). The bond between copper 2+ and water is a solid line, which implies a covalent rather than a complex bond. Structure c has some blue lines scribbled into the molecular structure. These look like covalent bonds with a long alkyl chain. Apparently, these should symbolize steric hindrance, which is very confusing because the same color and thickness is used as for the covalent bonds. Moreover, it is not clear from this picture where this supposed steric hindrance comes from.

 Response: The scheme was revised, as recommended.

- L 384: The water uptake values are practically identical considering the large error bars. Any conclusions on the chemical structures within the beads based on these values are void and mere speculation, unless the authors can demonstrate significant differences and a clear trend using statistical analysis.

Response: We thank the reviewer for this comment. The authors agree that the water uptake values may look very similar, however as seen from kinetic study results. The uptake studies may require longer equilibration time for the differences to become clearer. These and other observations are noted and will be part of extended studies that will focus on optimization of synthetic and adsorption conditions.

- L 410: Correlation with water uptake values are void (see comment above).

Response: We thank the reviewer for this comment. The authors agree that the water uptake values may look very similar, however as seen from kinetic study results. The uptake studies likely require longer contact time for the differences to become clearer. These and other observations are noted and will be part of further studies that will focus on optimization of synthetic and adsorption conditions.

- L 435: I see no experimental proof for greater accessibility here (or in the SEM images).

Response: We thank the reviewer for this comment. However, greater adsorption relates to the ability of the adsorbates to access surface or pore sites, which concur with the results presented herein.

- L 450: The presentation of these data is confusing and not uniform (e.g., different font sizes, non-uniform axis titles). The corresponding discussion of the data is questionable.

Response: The data as well as the discussion has been revised to address the reviewer query.

The graphs in figure 6A are very similar considering the error bar. A statistical test (which is missing) would probably reveal no statistical relevance, i.e., the swelling of all bead types is practically identical. The same can be assume for the date in 6D. The values for all beads are almost identical in most cases. Unfortunately, error bars are missing. Moreover, it is not clear here what “% removal” means (same goes for Qe and Qt. Figures (and table) need to be comprehendible without reference to the text.

What mathematical model(s) were use to fit curves to the scattered data points?

The SI unit for “minutes” is “min” (not “mins”).

Response: We thank the reviewer for the constructive comments. The authors agree that the water uptake values may look very similar, however as seen from kinetic study results. The uptake studies may require longer contact times for the differences to become clearer. These and other observations are noted and will be part of extended studies that will focus on optimization of synthetic and adsorption conditions.

% removal, Qt and Qe are not new, but are standard conventions that have been used in several publications

- L 455: As described above. Error bars and statistical analysis are needed. The values are very close together. Why does the removal efficiency increase in cycles 3 and 4? The differences are larger than the supposed differences between the different bead types. Moreover, I am not convinced that the whole process is sustainable considering that a lot of Cu2+ is used. How about leakage of cupper ions over time / the different cycles?

Response: We thank the reviewer for this comment. The focus of the study was to evaluate the effects of various modes of cross-linking on copper doping and adsorption properties of the beads. The authors agree that copper leaching may represent important parameter that will be considered as part of a future techno economic study planned for this project.

- L 494: I am not fully convinced that these absorbents are sustainable. The preparation requires immersion in an aqueous copper solution that needs to be disposed as heavy metal waste afterwards. Furthermore, the authors have not studied leaking of copper in the different cycles, which is basic requirement before application can be considered.

Response: We thank the reviewer for this comment. The focus of the study was to evaluate the effects of various modes of cross-linking on copper doping and adsorption properties of the beads. The authors agree that copper leaching may represent an important feature that will be explored in a techno economic follow-up study for this project..

- L 497: This claim is not justified and I cannot understand how the authors came to this conclusion. They studied ONE pH value. By comparison, most of the cited references reported a broader range of pH values.

Response: We thank the reviewer for this comment. The statement has been removed from the revised manuscript.

Summary: The authors appreciate the insightful and constructive comments provided by reviewer #4, along with the opportunity to improve this manuscript submission. The manuscript was further edited for language, syntax, and clarity to meet the high standards of this journal.

Round 2

Reviewer 3 Report

Comments and Suggestions for Authors

Detailed improvement has been made by the authors and therefore this revised version can be accepted for publication in the Journal in the current form. Thanks for the efforts from the authors.  

Author Response

Authors Response to Reviewer and Academic Editor’s comments on MS ID:  molecules-2767555

Reviewer #3 comments:

Detailed improvement has been made by the authors and therefore this revised version can be accepted for publication in the Journal in the current form. Thanks for the efforts from the authors.  

Response:  The authors appreciate the insightful and constructive comments provided by the reviewer #3. The manuscript has been checked for language, syntax, and clarity once again to meet the high standards of this journal.